# EdVAE: Mitigating Codebook Collapse with Evidential Discrete Variational Autoencoders

## Abstract

Codebook collapse is a common problem in training deep generative models with discrete representation spaces like Vector Quantized Variational Autoencoders (VQ-VAEs). We observe that the same problem arises for the alternatively designed discrete variational autoencoders (dVAEs) whose encoder directly learns a distribution over the codebook embeddings to represent the data. We hypothesize that using the softmax function to obtain a probability distribution causes the codebook collapse by assigning overconfident probabilities to the best matching codebook elements. In this paper, we propose a novel way to incorporate evidential deep learning (EDL) instead of softmax to combat the codebook collapse problem of dVAE. We evidentially monitor the significance of attaining the probability distribution over the codebook embeddings, in contrast to softmax usage. Our experiments using various datasets show that our model, called EdVAE, mitigates codebook collapse while improving the reconstruction performance, and enhances the codebook usage compared to dVAE and VQ-VAE based models.

## 1 Introduction

Generative modeling of images has been one of the popular research themes that aided in advancement of deep neural networks, particularly in enhancement of unsupervised learning models (Kingma & Welling, 2013; Goodfellow et al., 2014; Rezende & Mohamed, 2015; Ho et al., 2020; Rombach et al., 2021; Ramesh et al., 2021; 2022). Variational Autoencoders (VAEs) (Kingma & Welling, 2013) and their variant models have shown promise as solutions to numerous problems in generative modelling such as disentanglement of the representations (Higgins et al., 2017; Kim & Mnih, 2018), discretization of the representations (Jang et al., 2017; Maddison et al., 2017; Rolfe, 2016; Oord et al., 2017; Ramesh et al., 2021), and high-quality image generation (Vahdat & Kautz, 2020; Child, 2020; Razavi et al., 2019). Although most of the VAE models assume a continuous latent space to represent the data, discrete representations are more suitable to express categories that modulate the observation space (Oord et al., 2017). Supporting this rationale, recent celebrated large generative models also rely on discrete latent spaces learned by discrete VAEs to describe the image data (Esser et al., 2020; Rombach et al., 2021; Ramesh et al., 2021).

It is customary to form the latent space as a *codebook* consisting of discrete latent embeddings, where those embeddings are learned to represent the data. VQ-VAEs (Oord et al., 2017; Razavi et al., 2019; Esser et al., 2020) are discrete VAEs that quantize the encoded representation of the data by an encoder with the nearest latent embedding in the learnable codebook in a deterministic way. VQ-VAEs achieve considerably high reconstruction and generation performances. However, they are observed to suffer from the *codebook collapse* problem defined as the under-usage of the codebook embeddings, causing a redundancy in the codebook and limiting the expressive power of the generative model. As the deterministic quantization is the most likely cause of the codebook collapse problem in VQ-VAEs (Takida et al., 2022), probabilistic approaches (Takida et al., 2022), optimization changes (Huh et al., 2023) as well as codebook reset (Williams et al., 2020; Dhariwal et al., 2020) and hyperparameter tuning (Dhariwal et al., 2020) are employed in VQ-VAEs to combat the codebook collapse problem.

Unlike VQ-VAEs, the encoder of another discrete VAE, which is called dVAE (Ramesh et al., 2021), learns a *distribution* over the codebook embeddings for each latent in the representation. That means, instead of quantizing a latent with a single, deterministically selected codebook embedding, the

encoder of dVAE incorporates stochasticity to the selection of the embeddings where the learned distribution is modeled as a *Categorical* distribution. We find out that dVAE also suffers from the codebook collapse problem even though stochasticity is involved. One of our hypotheses is that attaining the probability masses for codebook embeddings using a softmax function induces a codebook collapse in dVAE, as we demonstrate in this paper.

Softmax notoriously overestimates the probability mass of the prediction, which in turn led to exploration of different alternatives to softmax, especially in classification tasks (Joo et al., 2020; Sensoy et al., 2018). Among those approaches, the widely adopted EDL (Sensoy et al., 2018) places a Dirichlet distribution whose concentration parameters over the class probabilities are learned by the encoder. In EDL, the class predictions are considered as the subjective opinions (Jøsang, 2016), and the evidences leading to those opinions are collected from the data, which are explicitly used as the concentration parameters of the Dirichlet distribution. To define such a framework, the softmax layer of the encoder is removed, and the logits of the encoder are used as the concentration parameters whose mean values are used as the predicted class probabilities. EDL can be also viewed as a generative model where the class labels follow a normal distribution whose mean is set by the uninformative Dirichlet prior over the class probabilities (Kandemir et al., 2022).

In this work, we find out that the root cause of the codebook collapse in dVAE can be framed as the artificial intelligence counterpart of the cognitive psychological phenomenon called the *confirmation bias* (Kahneman, 2011). This bias is developed cumulatively during the whole training process as the model overconfidently relates new observations to those already seen ones. We demonstrate by way of experiments that the spiky softmax probabilities lead to the confirmation bias problem. In order to mitigate the latter, we introduce an uncertainty-aware mechanism to map the inputs to the codebook embeddings by virtue of an evidential formulation. To that end, dVAE encoder collects evidences from the data, and the codebook embeddings that represent the data are selected based on those evidences. While the highest evidence increases the probability of the corresponding embedding to be selected, the collected evidences lead to a smoother probability distribution compared to softmax probabilities which leads to a diversified codebook usage. We reformulate the original EDL framework as a latent variable model for an unsupervised generative image modelling problem. We summarize our contributions as follows: (1) We introduce an original extension of dVAE with an evidential formulation. (2) We report evidence of the confirmation bias problem caused by the softmax probabilities used in dVAE. (3) We observe that the EDL integration improves the codebook usage of dVAE, which is measured by the perplexity.

In our work, we set the baseline results of dVAE for various datasets, and surpass the baseline measures with our model called Evidential dVAE (EdVAE) in terms of reconstruction performance, codebook usage, and image generation performance in most of the settings. We also compare EdVAE with state-of-the-art VQ-VAE based models using various experimental settings to demonstrate that our model performs close to or better than the VQ-VAE based methods.

## 2 RELATED WORK

**Codebook Collapse on VQ-VAEs** Vector quantization, which is useful for various tasks including image compression (Theis et al., 2017; Agustsson et al., 2017; Toderici et al., 2016), is the backbone of the VQ-VAE (Oord et al., 2017). While deterministically trained VQ-VAEs show favorable performance on image reconstruction and generation (Razavi et al., 2019; Esser et al., 2020), text decoding (Kaiser et al., 2018), music generation (Dhariwal et al., 2020), and motion generation (Siyao et al., 2022), some of the VQ-VAE variants use stochasticity and other tricks during the training, especially to alleviate the codebook collapse problem.

Codebook reset trick (Williams et al., 2020) replaces the unused codebook embeddings with the perturbed version of the most used codebook embedding during the training, and increases the number of embeddings used from the codebook. New perspectives on comprehending VQ-VAEs such as affine re-parameterization of the codebook embeddings, alternated optimization during the training, and synchronized update rule for the quantized representation are proposed by Huh et al. (2023) to address the problems of VQ-VAEs including the codebook collapse.

To incorporate stochasticity, a soft expectation maximization (EM) algorithm is reformulated based on the hard (EM) modeling of the vector quantization (Roy et al., 2018). GS-VQ-VAE (Sønderby

et al., 2017) uses the Euclidean distance between the encoder's output and the codebook as the parameters of a Categorical distribution, and the codebook embeddings are selected by sampling. SQ-VAE (Takida et al., 2022) defines stochastic quantization and dequantization processes which are parameterized by probability distributions. Those stochastic processes enable codebook usage implicitly without needing additional tricks such as codebook resetting.

Although the codebook collapse problem of the VQ-VAEs is studied in detail with observed success, it is still an open question for dVAEs. The dVAEs are employed instead of VQ-VAEs in (Ramesh et al., 2021) to obtain an image representation for the text-to-image generation problem. As the dVAEs have shown great potential in such complicated tasks, specifying existing problems of dVAE and providing relevant solutions are poised to bring both methodological and practical benefits.

To that end, our work proposes a novel way to combat the codebook collapse problem of the dVAEs. While the proposed methods for VQ-VAEs have the same objective of mitigating the codebook collapse problem like our method, our intuition differs since we focus on the properties of the distribution learned over the codebook, and try to attain a better distribution. On the other hand, other methods focus on the internal dynamics of the VQ-VAE model and its training.

**Evidential Deep Learning** The foundational work by Sensoy et al. (2018) has been pivotal in advancing uncertainty quantification in deep learning, and its methodology for modeling uncertainty inspires subsequent research. Soleimany et al. (2021) applies EDL to molecular property prediction, Bao et al. (2021) explores its use in open-set action recognition, and Wang et al. (2022) focuses on uncertainty estimation for stereo matching. These works collectively demonstrate the versatility and impact of the EDL framework, showcasing its influence across diverse domains. In our work, we employ EDL within a VAE-based framework for the first time to the best of our knowledge, marking an innovative incorporation of EDL.

## 3 BACKGROUND

### 3.1 DISCRETE VARIATIONAL AUTOENCODERS

Discrete VAEs aim to model the high-dimensional data $x$ with the low-dimensional and discrete latent representation $z$ by maximizing the ELBO of the log-likelihood of the data:

$$\mathcal{L}_{\text{ELBO}} = \mathbb{E}_{q(z|x)}[\log p(x|z)] - \text{KL}[q(z|x)||p(z)]. \tag{1}$$

where $p(x|z)$ is the generative model which is designed as a decoder, $q(z|x)$ is the approximated posterior which is designed as an encoder, and $p(z)$ is a prior.

In discrete VAEs, one common way to design the encoder output is to use a softmax head to learn a Categorical distribution over the possible values of discrete variables $z$. Then, the sampled discrete variable from the learned distribution is fed into the decoder to reconstruct the data $x$. As the discrete variables sampled from a Categorical distribution do not permit an end-to-end training, methods for continuous relaxation of the discrete variables are proposed, introducing the Gumbel-Softmax trick (Jang et al., 2017; Maddison et al., 2017).

VQ-VAE (Oord et al., 2017) is also a discrete VAE which uses vector quantization to construct a discrete latent space. In VQ-VAE, the latent variables are represented by a learnable codebook $\mathcal{M} \in \mathbb{R}^{K \times D}$, where $K$ is the number of embeddings, and $D$ is the dimensionality of each embedding. Each vector in the encoder's output $z_e(x) \in \mathbb{R}^{N \times N \times D}$ is quantized with the closest embedding in $\mathcal{M}$ using Euclidean distance, and the quantized representation $z_q(x) \in \mathbb{R}^{N \times N \times D}$ is obtained. $x$ is represented by $N \times N$ number of latent variables from the codebook, and $z_q(x)$ is fed into the decoder to reconstruct $x$.

Similar to VQ-VAEs, dVAE with a discrete encoder has been proposed by Ramesh et al. (2021). This method compresses the data $x$ into a latent tensor $z_e(x) \in \mathbb{R}^{N \times N \times K}$ which is taken as a distribution over the $K$ different codebook embeddings for each $N \times N$ spatial location. The training procedure of dVAE incorporates Gumbel-Softmax relaxation for differentiability of the sampled discrete variables.

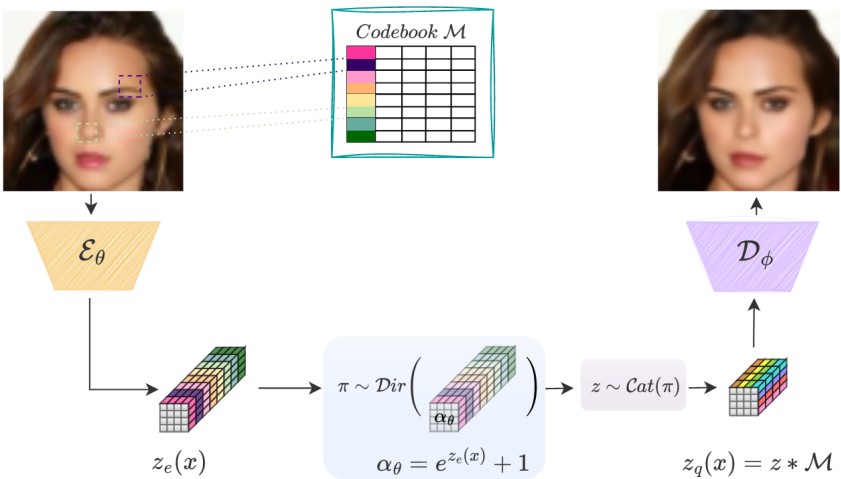

Figure 1: Overview of the method. An illustrative codebook is defined as $\mathcal{M} \in R^{8 \times 4}$ where 8 is the number of the codebook embeddings, 4 is the dimensionality of each embedding. For each 16 spatial positions in $z_e(x)$, we approximate a Dirichlet distribution as a distribution over the Categorical distributions that model the codebook embedding assignment to each spatial position.

## 3.2 EVIDENTIAL DEEP LEARNING

EDL enhances the Neural Networks (NNs) in order to allow for probabilistic uncertainty quantification for various tasks such as classification (Sensoy et al., 2018) or regression (Amini et al., 2020). As the deterministic NNs output a point estimate for the given input which does not comprise uncertainty over the decision, the EDL changes this behaviour such that the output of the NN is a set of probabilities that represent the likelihood of each possible outcome, by using the evidential logic (Dempster, 1968). Latent variable modeling of EDL (Kandemir et al., 2022) states the following generative model for the class labels of the observations:

$$\pi_n \sim \mathcal{D}ir(\pi_n | 1, \ldots, 1), \qquad y_n \sim \mathcal{N}(y_n | \pi_n, 0.5 I_K), \quad \forall n$$

where $\pi_n$ represents the latent variable of the observation $n$, and $K$-dimensional observations $y_n$, where $K$ is the number of classes, are represented with a multivariate normal prior. When the ELBO is derived, it is shown by Kandemir et al. (2022) that the negative ELBO is equal to the EDL loss.

## 4 METHOD

In this work, we enhance the dVAE model with an evidential perspective to mitigate the codebook collapse problem. To establish a stronger connection between the motivation and the proposed approach, it is essential to delve deeper into the root cause of codebook collapse. This issue arises when the model consistently relies on previously used codebook embeddings throughout training. Conceptually, the phenomenon of confirmation bias aligns with the codebook collapse problem. Over the course of training, the model gradually develops a bias, excessively associating new observations with previously encountered ones due to **overconfidence**. Given that the softmax function has been acknowledged to exhibit overconfidence issues (Joo et al., 2020; Sensoy et al., 2018), our motivation stems from the need to address the overconfidence challenge. Leveraging EDL, a framework demonstrated to alleviate the overconfidence problem associated with the softmax function, becomes a compelling solution. To sum up, overconfidence is the key connection between the evidential perspective and the codebook collapse problem, and EdVAE strategically leverages EDL's capabilities to counteract the codebook collapse.

Overview of the proposed method is shown in Figure 1. The encoder $\mathcal{E}_\theta$ outputs a tensor $z_e(x) \in R^{N \times N \times K}$ in agreement with the goal of learning a distribution over those embeddings. Originally, $z_e(x)$ in Figure 1 is passed to a softmax activation to obtain the probability masses $\pi$'s of the

Categorical distribution that models the codebook embedding assignment in dVAE. In EdVAE, we learn a distribution over the Categorical distributions as a Dirichlet distribution, and treat $z_e(x)$ as the concentration parameters $\alpha_\theta$ of this distribution. EDL approach builds an additional hierarchical level to the framework by its nature since the samples from Dirichlet distribution are also distributions. In Figure 1, the areas highlighted by blue and gray show the hierarchy of a distribution over a distribution. Unlike dVAE which has a single level of distribution as highlighted by gray, we employ an additional level in EdVAE highlighted by blue, in order to incorporate randomness with well quantified uncertainty over the selection process of the codebook embeddings.

EDL incorporation models a second-order uncertainty in EdVAE such that $\alpha_\theta$ represents how confident this prediction is, and $\pi$ predicts which codebook element can best reconstruct the input. The incorporated uncertainty awareness is expected to increase the codebook usage in EdVAE which is highly limited in dVAE due to the softmax operation. The intuition behind this expectation originates from the definition of the codebook collapse. New codebook elements are employed whenever the existing ones cannot explain the newcoming observation. Codebook collapse occurs when the model's prediction is wrong on whether the learned representations are capable of reconstructing the new observation or not. Therefore, employing uncertainty awareness over the codebook embedding selection enables the model to use unused codebook embeddings when it is uncertain about the codebook embedding selection. To validate this, we conduct an experiment revealing a correlation between uncertainty values and perplexity in CIFAR10, and this correlation supports our intuition (see Appendix E for details).

## 4.1 EdVAE Design

In order to place an evidential mechanism into dVAE model, the forward model and the design choices should be put in order. When we design our forward model, we follow a similar form to the latent variable modeling of EDL as described in Section 3.2, where now parameter $K$ represents the number of codebook embeddings. The concentration parameters, $\alpha_\theta$, of the Dirichlet distribution are defined to be greater than or equal to 1. Therefore, $z_e(x)$ is passed through an exp(.) operation to obtain the *evidences*, and 1 is added in order to obtain the concentration parameters as follows:

$$\alpha_\theta(x) = \exp(z_e(x)) + 1. \tag{2}$$

We define our forward model to be:

$$p(\pi) = \mathcal{D}ir(\pi|1,\ldots,1), \tag{3}$$
$$Pr(z|\pi) = \mathcal{C}at(z|\pi), \tag{4}$$
$$p(x|\mathcal{M}, z = k) = \mathcal{N}(x|\mathcal{D}_\phi(\mathcal{M}, z), \sigma^2 I). \tag{5}$$

In dVAE, the prior is defined as a uniform distribution over the codebook embeddings. Equation 3 demonstrates our prior design as a Dirichlet distribution that generates uniform distributions over the codebook embeddings on average, and $\pi = [\pi_1, \ldots, \pi_K]$. Equation 4 shows that we model the embedding selection as a Categorical distribution where $z$ is the index of the sampled codebook embedding from $\mathcal{M}$. During the training, we obtain samples from the Categorical distribution using Gumbel-Softmax (Jang et al., 2017; Maddison et al., 2017) relaxation to backpropagate gradients to the encoder. We decay the temperature parameter of the Gumbel-Softmax to 0 as described in (Jang et al., 2017) so that the soft quantization operation turns into the hard quantization. Therefore, there may be multiple embedding dimensions that are chosen independently due to the temperature value at train-time. At test-time, we perform hard quantization and simply take a single sample from Equation 4. The algorithms of the training and the inference of EdVAE are given in Appendix C. In Equation 5, $\mathcal{D}_\phi$ denotes the decoder network. The input of the decoder can be formed as $z_q(x) = z * \mathcal{M}$, which implies that the indices are used to retrieve the corresponding codebook embeddings (see Figure 1). The operator $*$ performs element-wise vector-matrix multiplication. We formulate the approximate posterior as:

$$q(\pi, z|x) = \mathcal{C}at(z|\pi)\mathcal{D}ir(\pi|\alpha_\theta^1(x), \ldots, \alpha_\theta^K(x)). \tag{6}$$

where $\alpha_\theta(x) = \left[\alpha_\theta^1(x), \ldots, \alpha_\theta^K(x)\right]$.

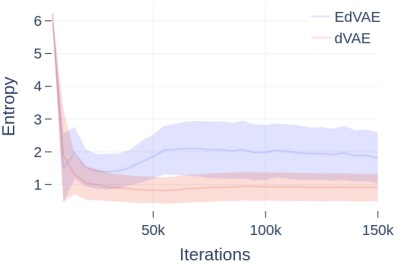
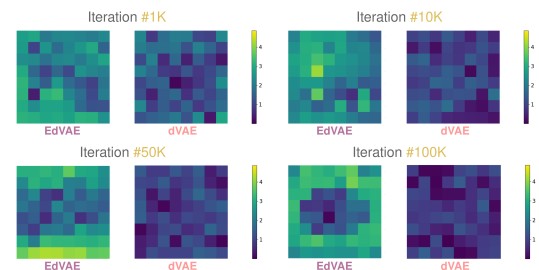

(a) Average entropy of the probabilities during the training.

(b) Entropy of the probabilities for each spatial position in the same sample during the training as heat maps.

Figure 2: Entropy visualization of the probability distributions for CIFAR10.

The ELBO to be maximized during the training is given by:

$$\mathcal{L}(\mathcal{M}, \theta, \phi) = \mathbb{E}_{Pr(z|\pi)} \left[ \mathbb{E}_{q(\pi|x)}[\log p(x|\mathcal{M}, z)] \right] - \mathcal{D}_{\text{KL}}(q(\pi|x)||p(\pi)). \tag{7}$$

The second term in Equation 7 is the Kullback-Leibler divergence between two Dirichlet distributions, hence has the analytical solution as derived in Appendix A. After further derivations over the first term in Equation 7 (Appendix B), our loss function is derived in Equation 8:

$$\mathcal{L}(\mathcal{M}, \theta, \phi) = \mathbb{E}_{q(\pi|x)} \left[ (x - \mathcal{D}_\phi(\mathcal{M}, z))^2 \right] - \beta \mathcal{D}_{\text{KL}}(Dir(\pi|\alpha_\theta(x))||Dir(\pi|1, ..., 1)) \tag{8}$$

which is optimized to increase the likelihood, while decreasing the KL distance regularized with $\beta$ coefficient (Higgins et al., 2017) between the amortized posterior and the prior.

The first term in Equation 8 indicates the reconstruction error in terms of mean squared error (MSE) between the input and the reconstruction. The objective goal of the model is to obtain a lower reconstruction error presuming that a well representing latent space is constructed. The second term indicates that the Dirichlet distribution which defines a distribution over the Categorical distributions should converge to a generated distribution in a uniform shape on average in order to represent the codebook embedding selection in a diverse way. When the distribution over the codebook embeddings is uniform, codebook usage is maximized as the probabilities of each codebook embedding to be selected become similar in value. Therefore, while the first term in Equation 8 aids directly in building the representation capacity of the latent space, the second term in Equation 8 indirectly supports that goal via a diversified codebook usage. We use a $\beta$ coefficient to balance the reconstruction performance and the codebook usage.

## 5 EXPERIMENTS

### 5.1 EFFECTS OF THE SOFTMAX DISTRIBUTION

Our hypothesis is that the spiky softmax distribution over the codebook embeddings develops confirmation bias, and the confirmation bias causes a codebook collapse. We test our hypothesis by comparing the average entropy of the probability distributions learned by the encoders of dVAE and EdVAE during the training using CIFAR10 dataset (Krizhevsky & Hinton, 2009). Low entropy indicates a spiky distribution while high entropy represents a distribution closer to a uniform shape, which is the desired case for a diverse codebook usage. A spiky probability distribution yields a confirmation bias because the codebook embedding with the highest probability mass is favorably selected. On the other hand, when the Categorical distribution has a flatter probability distribution, a variety of codebook embeddings might be sampled to represent the same data for its different details, which leads to an enriched codebook and an enhanced codebook usage.

Figure 2a visualizes the average entropy of the probabilities during the training. We measure the entropy of each probability distribution over the codebook embeddings sample-wise, meaning that

Table 1: Comparison of the models in terms of perplexity ($\uparrow$) using a codebook $\mathcal{M}^{512 \times 16}$.

| Method | CIFAR10 | CelebA | LSUN Church |
|---|---|---|---|
| VQ-VAE-EMA (Oord et al., 2017) | $412.67 \pm 2.05$ | $405.33 \pm 5.88$ | $379.67 \pm 3.09$ |
| GS-VQ-VAE (Sønderby et al., 2017) | $208.33 \pm 6.03$ | $193.33 \pm 10.68$ | $189.67 \pm 5.02$ |
| SQ-VAE (Takida et al., 2022) | $407.33 \pm 7.32$ | $\mathbf{409.33 \pm 2.05}$ | $374.00 \pm 2.16$ |
| VQ-STE++ (Huh et al., 2023) | $414.33 \pm 9.10$ | $370.33 \pm 4.11$ | $375.67 \pm 5.58$ |
| dVAE (Ramesh et al., 2021) | $190.33 \pm 13.02$ | $254.67 \pm 11.08$ | $363.33 \pm 4.07$ |
| EdVAE | $\mathbf{420.33 \pm 4.49}$ | $371.33 \pm 2.86$ | $\mathbf{385.67 \pm 5.63}$ |

Table 2: Comparison of the models in terms of MSE ($\times 10^3$, $\downarrow$) using a codebook $\mathcal{M}^{512 \times 16}$.

| Method | CIFAR10 | CelebA | LSUN Church |
|---|---|---|---|
| VQ-VAE-EMA (Oord et al., 2017) | $3.21 \pm 0.05$ | $1.07 \pm 0.06$ | $1.71 \pm 0.05$ |
| GS-VQ-VAE (Sønderby et al., 2017) | $3.63 \pm 0.01$ | $1.32 \pm 0.02$ | $1.84 \pm 0.06$ |
| SQ-VAE (Takida et al., 2022) | $4.01 \pm 0.03$ | $1.05 \pm 0.02$ | $1.79 \pm 0.03$ |
| VQ-STE++ (Huh et al., 2023) | $3.82 \pm 0.1$ | $1.11 \pm 0.08$ | $1.83 \pm 0.03$ |
| dVAE (Ramesh et al., 2021) | $3.42 \pm 0.08$ | $1.01 \pm 0.08$ | $1.60 \pm 0.01$ |
| EdVAE | $\mathbf{2.99 \pm 0.04}$ | $\mathbf{0.89 \pm 0.01}$ | $\mathbf{1.58 \pm 0.01}$ |

each sample consists of $N \times N$ number of entropy values calculated for each spatial position. We gather all $L \times N \times N$ entropy values at on the entire dataset where $L$ is the number of training samples, and plot the average entropy of the probabilities with mean and standard deviation. We find out that EdVAE's mean values of the entropy are higher than those of dVAE's, and this performance gain in terms of entropy is preserved during the training. Furthermore, we observe higher standard deviation for EdVAE in contrast to dVAE, which indicates that dVAE squeezes the probability masses into a smaller interval for all positions while the entropy values of EdVAE have a wider range aiding a relatively liberal codebook usage.

Figure 2b visualizes the entropy changes of the probabilities for each spatial position in the same sample during the training. All iterations have the same color bar for both models to observe the change effectively. Heat map visualization is useful to monitor the entropy of the probabilities within a single sample, and EdVAE obtains higher entropy values for most of the spatial positions compared to those of dVAE during the training.

We note that while the prior works (Williams et al., 2020; Takida et al., 2022) induce high entropy via regularizers, the feature introduced by our ELBO formulation inherently achieves the same effect as a result of our modeling assumptions that harness the power of the Dirichlet distribution.

While the spiky softmax distribution is the main problem, one might think to increase the stochasticity of sampling from the Categorical distribution with a higher temperature value to reduce the effects of overestimated probabilities. In order to test this assumption, we use higher temperature values with dVAE on CIFAR10 and CelebA datasets (see Appendix E for details). We show that the perplexity does not increase with a high temperature, and including randomness insensibly does not always affect the training in a good way. Our method incorporates stochasticity such that we learn how to perturb probabilities from the input. Learning a distribution over Categorical distributions from the input directly makes the model more reliable and resistant to hyperparameter change.

## 5.2 PERPLEXITY AND RECONSTRUCTION PERFORMANCE

We perform experiments on various datasets (Krizhevsky & Hinton, 2009; Liu et al., 2015; Yu et al., 2015) to demonstrate the performance of EdVAE compared to the baseline dVAE and VQ-VAE based methods. We use VQ-VAE-EMA (Oord et al., 2017) which updates the codebook embeddings with exponential moving averages, and GS-VQ-VAE (Sønderby et al., 2017) as the basic VQ-VAE models along with the state-of-the-art VQ-VAE based methods including SQ-VAE (Takida et al., 2022) and VQ-STE++ (Huh et al., 2023) mitigating codebook collapse problem.

In order to evaluate the codebook usage of all models, we leverage on the perplexity metric whose upper bound is equal to the number of the codebook embeddings. Therefore, we directly compare

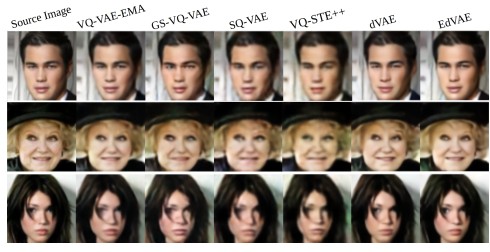 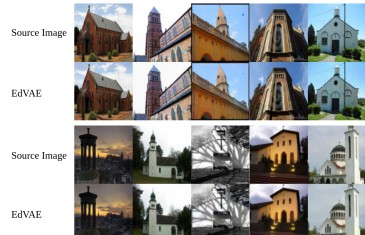

(a) Reconstructed samples from CelebA.       (b) Reconstructed samples from LSUN Church.

Figure 3: Reconstruction performance of EdVAE compared to other methods.

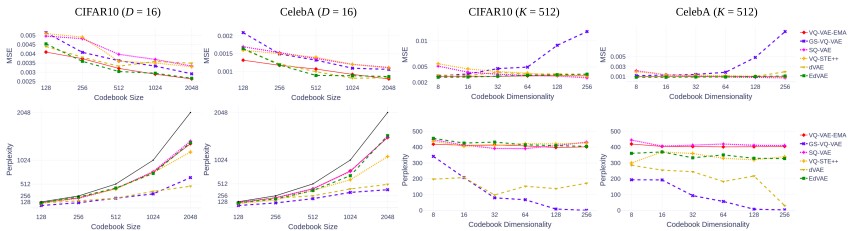

Figure 4: Impact of codebook design on perplexity and MSE, using CIFAR10 and CelebA datasets. The black "codebook size" line indicates the upper bound for the perplexity.

the number of used embeddings for all models. Perplexity results of all models are presented in Table 1. Additionally, since the reconstruction performance should not be lost while increasing the diversity of the codebook embeddings used in the latent representation, we also evaluate all models in terms of MSE. Numerical results are presented in Table 2. While we obtain the lowest MSE values for all datasets, we outperform the other methods for CIFAR10 and LSUN Church (Yu et al., 2015) datasets in terms of perplexity. Whereas SQ-VAE (Takida et al., 2022) achieves the highest perplexity result for CelebA (Liu et al., 2015) dataset, EdVAE performs close to SQ-VAE's perplexity, while obtaining a better reconstruction performance than SQ-VAE and the other methods. Moreover, EdVAE outperforms dVAE substantially in perplexity. It is important to note that EdVAE not only mitigates the codebook collapse problem of dVAE, but also outperforms the VQ-VAE based methods. All of the results are obtained after repeating the experiments with three different random seeds. Experimental details, dataset details, and architectural choices are further detailed in Appendix D.

We evaluate our model and the other models visually for a qualitative evaluation assessment. Figure 3a compares all of the models' reconstruction performance on CelebA dataset. We observe that while our model reconstructs most of the finer details such as the shape of eye or gaze direction better than the other models, we also include some examples where some of the finer details such as the shape of the nose and the mouth are depicted by one of the opponent models better than our model. We also add additional reconstructed images by EdVAE from the LSUN Church dataset in Figure 3b. Supplementary reconstruction results are given in Appendix E.

## 5.3 Effects of codebook design

We emphasize the critical role of codebook design using CIFAR10 and CelebA datasets in Figure 4. It is important to have a model that uses most of the codebook embeddings even with a larger codebook. Therefore, we evaluate EdVAE's and other methods' performance using various codebook sizes and dimensionalities. In order to observe the effects of size and dimensionality separately, we fix the dimensionality to 16 while we use different codebook sizes ranging from 128 to 2048. Then, we fix the size to 512 while we use different codebook dimensionalities ranging from 8 to 256.

Figure 4 shows that EdVAE outperforms the other methods in most of the setting where we use different codebook sizes. EdVAE's perplexity is not affected negatively when the codebook size increases, and it obtains the lowest MSE values in most of the settings for both of the datasets. Therefore, EdVAE is suitable to work with a larger codebook. When the codebook dimensionality

Table 3: Comparison of the models in terms of FID ($\downarrow$).

| Method | CIFAR10 | CelebA | LSUN Church |
|---|---|---|---|
| VQ-VAE-EMA (Oord et al., 2017) | $57.04 \pm 2.32$ | $34.30 \pm 2.41$ | $71.22 \pm 2.72$ |
| GS-VQ-VAE (Sønderby et al., 2017) | $56.35 \pm 2.17$ | $33.12 \pm 1.30$ | $72.52 \pm 3.25$ |
| SQ-VAE (Takida et al., 2022) | $54.17 \pm 2.85$ | $33.03 \pm 1.04$ | $\mathbf{63.41 \pm 2.36}$ |
| VQ-STE++ (Huh et al., 2023) | $55.53 \pm 1.97$ | $32.98 \pm 2.27$ | $71.03 \pm 1.95$ |
| dVAE (Ramesh et al., 2021) | $58.85 \pm 0.93$ | $37.29 \pm 3.14$ | $71.32 \pm 0.71$ |
| EdVAE | $\mathbf{51.82 \pm 1.58}$ | $\mathbf{32.51 \pm 1.13}$ | $69.63 \pm 1.29$ |

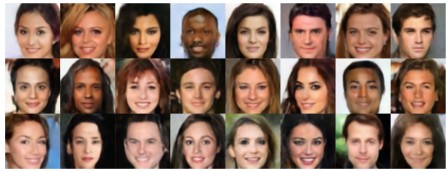

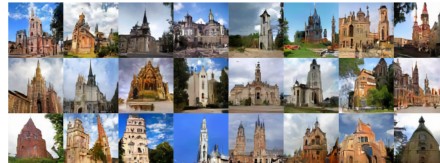

(a) Generated samples from CelebA.

(b) Generated samples from LSUN Church.

Figure 5: Generated samples using the learned prior over the discrete indices produced by EdVAE.

changes, we observe that EdVAE outperforms the other methods or obtains compatible results. While the other methods seem sensitive to the codebook design, EdVAE does not need a carefully designed codebook which eases the process of model construction.

### 5.4 APPROXIMATED PRIOR

Image generation is a downstream task over which we evaluate the performance of the discrete latent spaces learned by the discrete VAEs. As the prior used during the training of $\mathcal{E}_\theta$, $\mathcal{D}_\phi$, and $\mathcal{M}$ is a uniform distribution, i.e. an uninformative prior, it should be updated to accurately reflect the true distribution over the discrete latents. Therefore, we fit an autoregressive distribution over the discrete latents of the training samples, for which we follow PixelSNAIL (Chen et al., 2018). PixelSNAIL is also used in VQ-VAE-2 (Razavi et al., 2019) instead of PixelCNN (Oord et al., 2016) that is used in VQ-VAE (Oord et al., 2017) and SQ-VAE (Takida et al., 2022).

We utilize and report FID (Heusel et al., 2017) in Table 3 to evaluate the quality of the generated images that are created autoregressively after a training over the discrete codebook indices attained from a given model. The FID results imply that EdVAE performs better than the other models in all datasets except LSUN Church. These results elucidate the point that the discrete latent spaces learned by EdVAE helps to attain a reinforced representation capacity for the latent space, which is a desirable goal for the image generation task. Figure 5 presents generated samples by EdVAE whose discrete latent representations are produced by PixelSNAIL (see Appendix E for supplementary results).

## 6 CONCLUSION

The proposed EdVAE extends dVAE to mitigate the codebook collapse problem of the latter. We demonstrate the essence of the problem that is the confirmation bias caused by the spiky softmax distribution, and reformulate the optimization with an evidential view in order to acquire a probability distribution aiding codebook usage. We show that our method outperforms the former methods in most of the settings in terms of reconstruction and codebook usage metrics.

Although the proposed method improves the codebook usage for dVAE family, there is still room for improvements both numerically and experimentally. We evaluate our method over small to medium size datasets compared to datasets like ImageNet (Deng et al., 2009). As our work is the first to state the codebook collapse problem of dVAE and to propose a solution to it, it sets a baseline that can be extended to obtain improved outcomes in different models and datasets. We mainly provide evidence for the utility of our method for relatively structured datasets like CelebA or LSUN Church, and exploring the best parameterization for more diverse datasets like ImageNet would be a possible direction of our future work.

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
