# OpenReview forum: "EdVAE: Mitigating Codebook Collapse with Evidential Discrete Variational Autoencoders"
_ICLR.cc/2024/Conference — ICLR 2024 Conference Withdrawn Submission_

### Official Review · Reviewer_m7N7 · 2023-10-31

**Soundness:** 3 good
**Presentation:** 2 fair
**Contribution:** 3 good
**Rating:** 5
**Confidence:** 3

**Summary:**

Targeting at the codebook collapse problem in training deep generative models, this work incorporates evidential deep learning instead of softmax to mitigate the overconfidence issue in matching codebook elements. Experiments on various datasets show that the proposed EdVAE method mitigates codebook collapse and improves the reconstruction performance. Source codes are provided in the supplementary material.

**Strengths:**

+ The motivation for using evidential deep learning to mitigate the overconfidence brought by Softmax is clear and convincing.
+ The paper is well-written and detailed math derivations are provided in the supplementary material, which makes theoretical contributions.
+ Comprehensive experiments are conducted, which demonstrate the effectiveness of this method.
+ Source codes are provided, which makes this work easy to reproduce.

**Weaknesses:**

- The Related Work section lacks works on evidential deep learning.
- It seems that the coefficient \beta in Eq.(8) is set to different values for different methods on different datasets. Is the experiment performance very sensitive to this coefficient? A hyper-parameter analysis experiment on \beta may be useful for better evaluating the effectiveness of the method.
- In the EDL formulation, uncertainty is calculated by u=C/S, which indicates that the uncertainty is inversely proportional to the sum of evidence. However, in your method, there is no explicit use of uncertainty. What's the reason?
- (Minor) How to ensure the evidence learned from the codebook meaningful?

**Questions:**

See weaknesses.

---

> ### Author Response · Authors · 2023-11-16
> **Answers to Reviewer m7N7's Questions**
>
> We thank reviewer for comments and questions.
>
> (1) We added works on Evidential Deep Learning to the Related Work section. Please see the updated text.
>
> (2) Considering the reviewer’s suggestion, we acknowledge the importance of this parameter in our method and have conducted additional experiments to analyze its impact on performance (see Appendix E.4). We also corrected our mistake in Appendix D as we actually did not use different beta coefficients for different datasets in dVAE. We thank the reviewer for helping us realize this.
>
> To start with, we stated that based on the model and the dataset, the upper bound for the beta coefficient varies in our submission. The reason is that the KL terms are different in different models, and the beta coefficient decides the reconstruction vs KL term tradeoff. Therefore, it is natural to use different beta coefficients for different methods.
>
> For the effects of beta coefficient in EdVAe, we performed several experiments by changing the beta coefficient within the 1e-7 - 1e-4 range. We repeated our experiments for dVAE and EdVAE using all of the datasets. We observed that our method is more sensitive to beta coefficient than dVAE, and EdVAE diverges when the beta coefficient increases for all datasets. We think that the key factor to this sensitivity is the complexity introduced by the KL distance between our newly introduced posterior and prior, compared to the KL distance in dVAE. Therefore, fine-tuning the beta coefficient emerges. Even though our original KL term brings some sensitivity to the training and it requires a hyper-parameter tuning like most of the AI models, its contribution to the performance is non-negligible and essential.
>
> Besides, the best performing beta coefficient for CIFAR10 dataset is slightly higher than the best performing beta coefficient of CelebA and LSUN Church datasets. Our intuition for this difference is that, reconstructing images with lower resolution as in CIFAR10 is less challenging than reconstructing images with higher resolution as in CelebA and LSUN Church. Therefore, increasing the beta coefficient from 1e-7 to 5e-7 improves the performance in CIFAR10 without hurting the reconstruction vs KL term tradeoff. On the other hand, 1e-7 to 5e-7 conversion slightly decreases the performance in CelebA and LSUN Church datasets since the reconstruction of the higher resolution images affects the reconstruction vs KL term tradeoff.
>
> (3) Our intuition in EdVAE is that uncertainty awareness will enhance codebook usage since new elements will be introduced when existing ones fail to explain the observations, which indicates the uncertainty of the model. Thanks to the reviewer’s question, we conducted an experiment to validate our intuition, and shared our results in Appendix E.2.
>
> We observed the expected correlation between the perplexity and the uncertainty results during the training of EdVAE in CIFAR10: perplexity values increase during the training due to the increase of uncertainty values. This result emphasizes that our model dynamically adjusts codebook usage based on its uncertainty.
>
> As the uncertainty already affects the training implicitly to support high perplexity values, there is no reason to regularize the training by the explicit usage of uncertainty.
>
> (minor) The evidence is learned from the data by the encoder. In dVAE, the encoder learns logits from the data as a distribution over the codebook, and the probabilities of this distribution is calculated using softmax. In EdVAE, we use the encoder’s output as the evidence to learn a distribution over the codebook. Therefore, we do not learn evidence from the codebook. As the evidence is learned to construct a distribution that can fully cover the codebook, the only way to measure the meaningfulness of the evidence is to measure perplexity which depicts the usage of the codebook embeddings. If the perplexity is high, we can infer the effectiveness of the evidence. Besides, the reconstruction performance is also important to evaluate the effects of evidence learning as lower reconstruction loss indicates trustworthy evidence.

---

> > ### Comment · Reviewer_m7N7 · 2023-11-21
> > **Thank you for your response**
> >
> > Thank you for your detailed answers to the other reviewers and me. I still don't very understand why the uncertainty cannot be used explicitly in the proposed method (whereas many others do, for example, [a-b]).  Although I do not have strongly negative opinions about this paper, I keep my original score.
> >
> > [a] Evidential deep learning for open-set action recognition. ICCV, 2021.
> > [b] Dual-evidential learning for weakly-supervised temporal action localization. ECCV, 2022.

---

### Official Review · Reviewer_QdFK · 2023-11-01

**Soundness:** 2 fair
**Presentation:** 3 good
**Contribution:** 2 fair
**Rating:** 5
**Confidence:** 5

**Summary:**

The paper presents a novel approach, Evidential Discrete Variational Autoencoders (EdVAE), designed to mitigate the codebook collapse issue observed in Variational Autoencoders (VAEs). To address the confirmation bias issue associated with discrete VAEs (dVAE), the authors replace the softmax function with evidential deep learning (EDL). Extensive experimental results are provided to substantiate the effectiveness of EdVAEs in comparison to dVAEs.

**Strengths:**

- The authors adeptly pinpoint and tackle the confirmation bias problem induced by the softmax probabilities in dVAEs.
- Comprehensive experimental results are showcased, underlining the efficacy of the proposed method against established benchmarks.
- The paper is articulated well, featuring clear explanations and a coherent structure.

**Weaknesses:**

- There is a discrepancy between the method outlined in the text and its code implementation. The paper describes a two-stage sampling process: first sampling a distribution over the codebook from a Dirichlet distribution, parameterized by the output logits, and then sampling a code from this distribution. However, the code implementation directly samples the code from a categorical distribution parameterized by the output logits. Given this, it is unclear why EdVAE, which optimizes a more complex expression for KL divergence, outperforms dVAE, which directly optimizes entropy, when higher entropy is the desired outcome.
- Beyond the computation of the KL divergence, there are two other distinctions between EdVAE and dVAE: 1. The value of \alpha^i
  is capped at a maximum of 20, and 2. EdVAE utilizes a fully connected (FC) layer to compute logits instead of calculating the distance to each code embedding. It would be beneficial to explore whether these modifications would enhance dVAE's performance as well.
- The experimental results could be more compelling. The paper primarily offers quantitative results, but the improvements in MSE and FID over other methods appear to be marginal.

**Questions:**

- Can the authors clarify the inconsistency between the described method and the code implementation?
- Could the authors elucidate why EdVAE is more effective, given that the goal is to achieve higher entropy, even though EdVAE optimizes a different metric?
- Could the authors conduct ablation studies on alpha clamping and logits computation?

---

> ### Author Response · Authors · 2023-11-16
> **Answers to Reviewer QdFK's Questions**
>
> We thank reviewer for the comments and detailed analysis.
>
> (1) In our implementation, we used a common approach to stabilize the training that is also used in EDL (Sensoy et al., 2018) paper and its implementation [1]. As stated in Equation 2 of (Sensoy et al., 2018), the expected probability for the kth singleton is the mean of the corresponding Dirichlet distribution as p_k = α_k / ∑α. Therefore, we used the expected probabilities as the probabilities of the Categorical distribution instead of sampling from the Dirichlet distribution. This approach is commonly used in evidential methods’ implementations (see line 44 of [1] and line 90 of [2]) Additionally, double sampling causes an instability in the training. The mean of the Dirichlet distribution that we calculated in the 34th line of dirichlet_quantizer.py can be used theoretically and practically. Therefore, there is not an inconsistency between the method and the code implementation. We also explained our approach to attain probabilities πs in Appendix B. Please see the updated text for further explanation.
> [1] https://github.com/dougbrion/pytorch-classification-uncertainty/blob/master/losses.py
> [2] https://github.com/ituvisionlab/EvidentialTuringProcess/blob/main/etp.py
>
> (2) As we stated in Section 5.1, there are some methods (Williams et al., 2020; Takida et al., 2022b) which explicitly use entropy as regularizers. However, we want to clarify that neither dVAE nor EdVAE directly optimizes entropy. EdVAE inherently achieves higher entropy compared to dVAE because of the stochasticity that we managed to involve systematically. In our design, the prior is modeled as a Dirichlet distribution that generates uniform distributions over the codebook embeddings on average, which includes stochasticity. On the other hand, dVAE’s prior is a deterministic uniform distribution. Therefore, our KL term brings stochasticity to the training which eventually leads to higher entropy.
>
> Not only our KL term, but also learning a distribution over distribution with an evidential perspective increases the stochasticity. We also analyzed the effects of temperature parameter in dVAE which affects the stochasticity in Appendix E.1, and showed that systematically incorporating stochasticity to the training as in EdVAE leads to higher and more stable performance compared to dVAE.
>
> (3) For the logit computation part, we would like to clarify that both dVAE and EdVAE compute logits with the same way: the logits are the output of a convolutional (Conv2D) projection layer whose channel size is equal to the number of embeddings in the codebook (see our implementation dvae_encoder.py, line 42) since both dVAE and EdVAE learns logits as a distribution over the codebook embeddings. Therefore, this way of logit calculation is not EdVAE’s proposal. On the other hand, logits of a Categorical distribution are calculated as the distances to each codebook embedding in GS-VQ-VAE as we stated in the Related Work section. Therefore, the Reviewer’s suggestion to explore the effects of logit calculation can be already achieved by comparing GS-VQ-VAE and dVAE.
>
> For the alpha clamping part, we followed this common approach of logit clamping to stabilize the training since the exponential of logits might be really large (see Line 110 in [2]). Considering the Reviewer’s suggestion, we conducted an ablation study to observe the effects of alpha values (see Appendix E.3). We observed: (1) Clamping the logits with smaller max values clamps some of the values in logits, and limits the range of positive values logits can have. This situation limits the representativeness of the logits. (2) Using larger max values for clamping causes instability in the training as the exponential of logits gets large. Therefore, the logits should be clamped eventually with proper values. (3) If a proper max value can be selected, clamping acts as a regularizer at the beginning of the training, and the encoder naturally outputs logits with no values greater than the max clamping value after a few iterations. (4) If the training is already stabilized, the max clamping value does not affect the performance dramatically. Therefore, using 20 as the max value for all datasets can be a mutual design choice.

---

### Official Review · Reviewer_xKyk · 2023-11-02

**Soundness:** 2 fair
**Presentation:** 3 good
**Contribution:** 2 fair
**Rating:** 3
**Confidence:** 4

**Summary:**

In this paper, the authors proposed an original extension of discrete VAE with an evidential formulation (EdVAE) to tackle the problem of Codebook collapse. To be Specific, the authors utilize a Dirichlet instead of distribution as a distribution instead of stochastic process over the Categorical distributions that model the codebook embedding assignment to each spatial position. Extensive experiments demonstrate that the proposed methord improves the current benchmark performance.

**Strengths:**

(1)	The paper is organized and clearly written.
(2)	In this paper, the author attempted to utilize Dirichlet distribution to solve the problem of Codebook Collapse caused by stochasticity, which seems to be intuitively reasonable.
(3)	Sufficient experimental results demonstrate the effectiveness of the proposed method.

**Weaknesses:**

(1)	The proposed method lacks innovation. The authors proposed to utilized Evidential Deep Learning (EDL) to tackle the problems of codebook collapse that are the combination of two proposed framework.
(2)	There are few recently proposed methods in the experimental results so that I do not know whether the proposed method achieves the superior performance nowadays.
(3)	The paper lacks ablation experiments, which cannot prove the effectiveness of the proposed module.
(4)	The Motivation is unclear. The authors proposed to tackle the Codebook collapse problem in this paper. However, the proposed method has little relevance to the motivation. I would appreciate it if the authors could further explain the rationale for the proposed approach.

**Questions:**

Please see the Weaknesses.

---

> ### Author Response · Authors · 2023-11-11
> **Ablation Experiments**
>
> We thank Reviewer xKyk for the comments and questions. In response to Weakness 3, we kindly want to ask what ablation experiments that the reviewer requests. Our method EdVAE adds an additional modular stage to baseline dVAE, and we have already compared the effects of our solution by comparing EdVAE with the baseline dVAE both qualitatively and quantitavely. If the reviewer is specifically interested in the effects of a particular hyperparameter as part of an ablation study or has any other specific requests, we would be happy to perform additional experiments to address their concerns.

---

> ### Author Response · Authors · 2023-11-16
> **Answers to Reviewer xKyk's Questions**
>
> We thank reviewer for helping us improve our work.
>
> (1) While it may seem that our method utilizes EDL in a straightforward manner, our integration of EDL is not a mere application but rather a novel adaptation. We have modified and extended the EDL framework to be applicable in the context of VQ-VAE. To the best of our knowledge, this is the first instance in the literature where EDL has been tailored for use within a VAE framework. This adaptation involves using the Dirichlet distribution as a distribution over Categorical distribution which is the first in the literature. The derivation details in Appendix B further emphasize our contribution. Moreover, our work addresses the issue of codebook collapse, combining EDL with another proposed framework, which collectively provides a unique and effective solution to the problem. We believe that the integration of these two components and their application to address codebook collapse is a distinctive contribution to the existing literature.
>
> (2) In our experimental evaluation, we performed a comprehensive comparison with the most recent state-of-the-art method, VQ-STE++. To the best of our knowledge, VQ-STE++ is currently considered one of the leading approaches in the field. Our decision to compare against VQ-STE++ was based on its recent introduction and its performance superiority over existing methods.
>
> Additionally, we included a comparison with the commonly used method SQ-VAE to provide a broader context for the evaluation. It's worth noting that VQ-STE++ itself compares against SQ-VAE. Thus, our choice of baseline models aligns with the current state-of-the-art landscape in the field.
>
> While our baseline model is dVAE, not VQ-VAE, this choice was deliberate as we aimed to address the common challenge of codebook collapse, which is pertinent to both dVAE and VQ-VAE based methods. This ensures a comprehensive evaluation of our proposed method against a range of relevant approaches.
>
> (3) We have included an ablation experiment to investigate the impact of the logits clamping to obtain the alpha parameters. This experiment aims to provide insights into the sensitivity of our proposed method to variations in this parameter and assess its influence on the overall performance (Appendix E.3). We observed that clamping is important to regularize the training and our method is properly optimized.
>
> Furthermore, we have included an ablation experiment focusing on the beta parameter in the KL term. This analysis is designed to elucidate the contribution of the beta parameter to the model's performance, offering a more granular understanding of its role in the proposed framework (Appendix E.4). We observed that beta coefficient plays a critical role for the performance of our method since the KL distance between our newly introduced posterior and prior is more complex than the original KL distance in dVAE. Even though our original KL term brings some sensitivity to the training, its contribution to the performance is essential.
>
> Lastly, we conducted an experiment to see if the uncertainty value encourages the perplexity. We observed that perplexity values increase due to the increase of uncertainty values which validate our intuition for incorporating uncertainty awareness to the model (Appendix E.2).
> These experiments have been conducted with the intention of providing a more comprehensive evaluation of our proposed method. We believe that the results of these experiments contribute valuable insights into the robustness and effectiveness of our approach under different parameter configurations.
>
> (4) The codebook collapse problem, wherein the model tends to reuse embeddings during training, is linked to confirmation bias. Confirmation bias occurs due to overconfidence, and the overconfidence issue arises with the softmax function usage as stated in the literature. We experimentally observed the entropy of the probability values that are obtained using the softmax function. We noticed that the entropy of the distribution is not high enough to support stochasticity, indicating that a few of the codebook embeddings have overestimated probabilities. That is why, we thought about how we can incorporate solutions that are useful to solve the overestimation problem of the softmax. Overconfidence is the key connection between the evidential perspective and the codebook collapse problem: EDL is developed to mitigate the problems of the softmax function, and our motivation is to mitigate the softmax function causing the codebook collapse problem. Therefore, EdVAE incorporates EDL into dVAE to strategically address codebook collapse. We  updated the 1st paragraph of Method for a more detailed explanation of our motivation.

---

### Author Response · Authors · 2023-11-16
**General Comments**

We thank all reviewers for helping us improve our work. We answer all of their questions and concerns separately since each of them focus on different aspects of our method. Thanks to all reviewers comments and questions:

1. We added new experiments for beta coefficient, logits clamping, and uncertainty effects on perplexity. Please see our updated Appendix.
2. We clarify our explanations for better understanding. Please see our updated text.
3. We extended Related Work section, and included EDL related works.

We would be happy to discuss our rebuttal with the reviewers, and answer their further questions.